# A pipeline for malignancy and therapy agnostic assessment of cancer drug response using cell mass measurements

Robert J. Kimmerling [1,11✉], Mark M. Stevens[1,11], Selim Olcum[1,11], Anthony Minnah[1,12], Madeleine Vacha[1,12], Rachel LaBella[1,12], Matthew Ferri [1], Steven C. Wasserman[1], Juanita Fujii[2], Zayna Shaheen[2], Srividya Sundaresan[2], Drew Ribadeneyra[3], David S. Jayabalan[3], Sarita Agte[4,5], Adolfo Aleman[4,5,6], Joseph A. Criscitiello[7], Ruben Niesvizky[3], Marlise R. Luskin [7], Samir Parekh [4,5,8,9], Cara A. Rosenbaum[3], Anobel Tamrazi[10] & Clifford A. Reid [1✉]

Functional precision medicine offers a promising complement to genomics-based cancer therapy guidance by testing drug efficacy directly on a patient's tumor cells. Here, we describe a workflow that utilizes single-cell mass measurements with inline brightfield imaging and machine-learning based image classification to broaden the clinical utility of such functional testing for cancer. Using these image-curated mass measurements, we characterize mass response signals for 60 different drugs with various mechanisms of action across twelve different cell types, demonstrating an improved ability to detect response for several slow acting drugs as compared with standard cell viability assays. Furthermore, we use this workflow to assess drug responses for various primary tumor specimen formats including blood, bone marrow, fine needle aspirates (FNA), and malignant fluids, all with reports generated within two days and with results consistent with patient clinical responses. The combination of high-resolution measurement, broad drug and malignancy applicability, and rapid return of results offered by this workflow suggests that it is well-suited to performing clinically relevant functional assessment of cancer drug response.

[1] Travera, Medford, MA, USA. [2] Department of Clinical Research, Dignity Health, Sequoia Hospital, Redwood City, CA, USA. [3] Weill Cornell Medicine, New York, NY, USA. [4] Department of Medicine, Hematology and Medical Oncology, Icahn School of Medicine at Mount Sinai, New York, NY, USA. [5] Tisch Cancer Institute, Icahn School of Medicine at Mount Sinai, New York, NY, USA. [6] Graduate School of Biomedical Sciences, Icahn School of Medicine at Mount Sinai, New York, NY, USA. [7] Department of Medical Oncology, Dana-Farber Cancer Institute, Boston, MA, USA. [8] Precision Immunology Institute, Icahn School of Medicine at Mount Sinai, New York, NY, USA. [9] Department of Oncological Sciences, Icahn School of Medicine at Mount Sinai, New York, NY, USA. [10] Division of Vascular and Interventional Radiology, Palo Alto Medical Foundation, Redwood City, CA, USA. [11]These authors contributed equally: Robert J. Kimmerling, Mark M. Stevens, Selim Olcum. [12]These authors contributed equally: Anthony Minnah, Madeleine Vacha, Rachel LaBella. ✉email: rkimmerling@travera.com; creid@travera.com

Effective biomarkers for precision oncology require measurement approaches that, in addition to predicting a patient's response to therapy, are amenable to data collection within the constraints of routine clinical cancer care. Key constraints include limited amounts of tumor specimens for characterization, biological heterogeneity, and the requirement for rapid return of results to ensure clinical actionability.

Genomic biomarkers have become the gold standard for guiding therapy choice in precision oncology, demonstrating remarkable clinical benefit for several well-defined genomic alterations[1–3]. However, recent clinical results have shown that such genomics-driven approaches remain limited in scope. Most notably, the National Cancer Institute—Molecular Analysis for Therapy Choice (NCI-MATCH) study found that fewer than 20% of patients were assigned to a therapy based on the identification of an actionable mutation[4]. When also considering patient outcomes, recent studies have found that approximately 5–7% of patients demonstrate clinical benefit from genome-targeted therapies[5,6]. Furthermore, patients that initially respond to treatment often develop resistance, at which point subsequent analysis of actionable mutations rarely offers additional information to guide further treatment[7]. Thus, there is a critical need for new tools that complement genomic approaches to enable rational therapy selection for a wider population of cancer patients.

Functional precision medicine offers one such approach[8,9]. Whereas molecular, histological, and genomic biomarkers for predicting cancer drug response rely on proxy measurements of cellular function to potentially inform drug selection, functional precision medicine instead measures the effect of specific drugs directly on live cells isolated from a patient's tumor. This approach has the benefit of offering a truly personalized biomarker for potential drug efficacy. However, the need for live cells presents distinct challenges to testing, including limited cellularity offered by the most clinically accessible specimen formats, loss of cell viability, and rapid phenotypic drift ex vivo. Because of these constraints, many recent development efforts in functional precision medicine have focused on hematologic malignancies where access to an abundance of live cells from a fresh tumor specimen is more routinely feasible[10–15]. However, broader application of these approaches to solid tumors remains challenging, often requiring extended culture to expand cells ex vivo to enable drug response testing[16–18]. Despite encouraging recent progress towards testing drugs more quickly on freshly isolated solid tumor cells[19], these biomarkers have yet to be translated into workflows that enable routine clinical testing.

Key requirements for a broadly applicable functional test for cancer drug response include (1) Flexibility: the test must be compatible with various tumor specimen formats that are collected as part of standard clinical care and the biomarker must demonstrate an ability to detect response to drugs of different classes. (2) Sensitivity and robustness: the assay must be capable of identifying subtle changes in cell populations from highly heterogenous specimens. (3) Speed: loss of cell viability and rapid phenotypic drift ex vivo often preclude long-term drug dosing strategies upstream of response assessment. Furthermore, to effectively guide therapeutic decision-making, particularly for patients with advanced or progressing diseases, the assay results must be returned in a clinically actionable timeframe.

Single-cell mass measurements are uniquely suited to meet the translational requirements for functional precision medicine assays. As an integrative biophysical readout of phenotype, cell mass has been shown to change rapidly in response to treatment with efficacious drugs, and, as a single-cell measurement, populations can be characterized using a relatively small number of cells[20–24]. Additionally, these measurements of cellular mass change have been shown to correlate with patient treatment response in a range of malignancies[23,24]. However, as with other functional measurement approaches, the logistical and technical challenges of performing these live cell measurements continue to limit their clinical applicability.

Here we describe an end-to-end workflow that extends the potential application space of single-cell mass-based drug response testing by demonstrating the feasibility of using this approach to assess drug efficacy in various primary specimen formats from both hematologic and solid tumor malignancies (Fig. 1). Using shipping and tumor cell isolation protocols specific for various specimen formats, viable single-cell suspensions are generated (Methods). After overnight drug treatment in vitro, the mass distributions of these cell populations are measured using a microfluidic platform that incorporates a suspended microchannel resonator (SMR). The SMR sensor enables highly precise

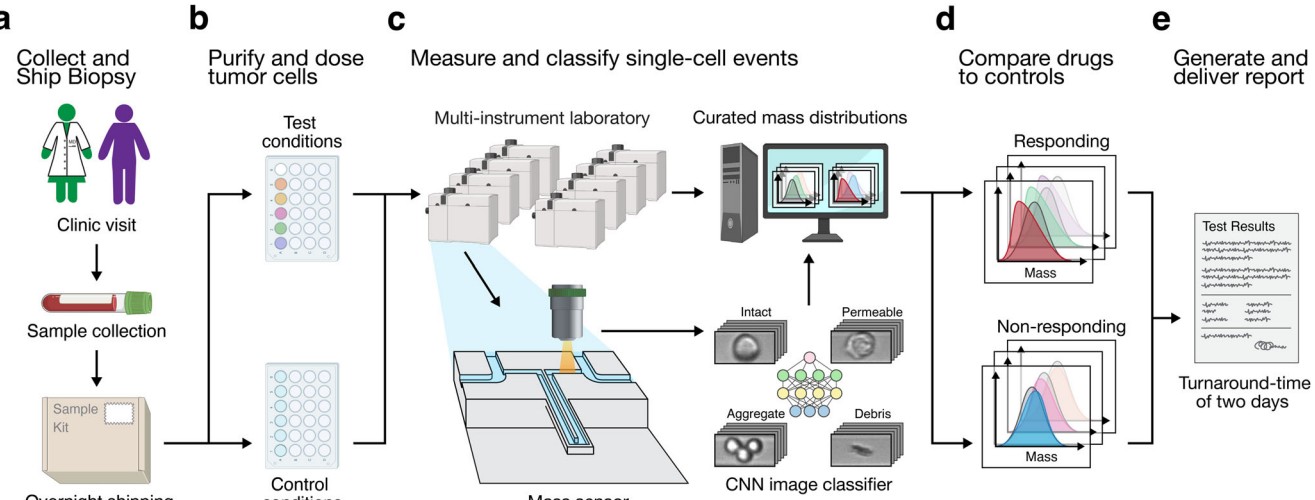

**Fig. 1 Clinical pipeline for cancer drug responsiveness testing using image-curated single-cell mass measurements.** Schematic overview of the full drug testing pipeline including **a** sample collection and shipment, **b** tumor cell isolation and overnight drug incubation, **c** single-cell mass data collection using more than twelve SMR-based instruments with linked brightfield imaging, downstream image classification, **d** statistical analysis, **e** response calling and results reporting.

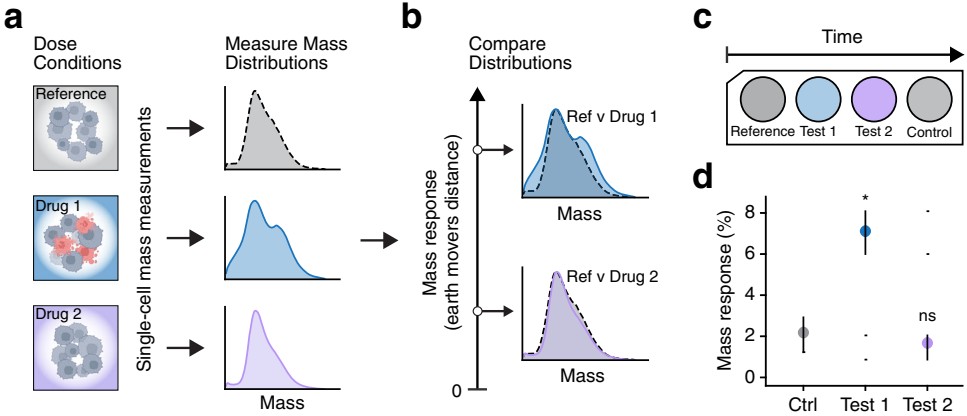

**Fig. 2 Measuring the mass response of cancer cells to treatment. a** First, isolated cancer cells are dosed and incubated with drugs to be tested (blue and purple) or vehicle-treated control (gray). The mass of individual cells from each population are measured by the SMR instrument to determine the mass distribution of treated cells (blue and purple curves) and untreated reference cells (gray curve). **b** Mass distributions of the treated and reference cells are compared to determine the mass response to treatment. The mass response signal is the statistical distance between the two mass distributions calculated by the Earth Mover's Distance (Methods). Comparing two similar distributions (purple vs gray) yield a smaller mass response signal, whereas comparing diverging distributions (blue vs gray) results in a larger mass response signal. **c** Schematic representation showing the structure of the mass response test. Vehicle-treated control cells are measured both before and after the cells that are treated with the drugs being tested. This approach enables measuring a baseline mass response signal between two control populations (reference vs control), which captures possible time-varying mass changes due to in vitro effects. **d** Mass response plot showing the mass response signals of the control and treated cells shown in (**a**) and (**b**). 95% confidence intervals, shown as lines overlaid on each point, are calculated by bootstrapping using single-cell mass data (Methods). The *p* values are determined by comparing the mass response magnitude difference of TEST and CTRL to a "limit of decision" threshold of 3% (Methods). * indicates *p* < 0.05, ns indicates *p* > 0.05.

measurements of cell mass in a simple flow-through channel configuration, as described extensively elsewhere[25–27]. In addition to mass measurements, brightfield images are collected for each particle passing through the mass sensor and subsequently annotated using a convolutional neural network (CNN)-based image classifier to ensure that only single cells of interest are used for downstream statistical analysis. This complete workflow demonstrates robust technical reproducibility (Supplementary Fig. 1) and is routinely completed within 2 days.

Using this pipeline, a wide range of therapies can be tested within 24 h of treatment, as demonstrated here, with dose-dependent mass responses for 60 different drugs with varying mechanisms of action across various cell lines. Furthermore, we demonstrate the feasibility of performing these measurements for several different minimally-invasive tumor specimen formats that are commonly collected as part of routine clinical care. These include blood, bone marrow, low-input specimens such as fine-needle aspirates (FNA), and malignant fluids such as pleural effusions. Together, these results demonstrate that a test based on image-annotated single-cell mass measurements has the potential to offer broad utility, is robust to the biological complexity of translationally-relevant clinical specimens, and can be executed in a clinically actionable timeframe.

## Results

**Measuring mass response signal by comparing single-cell mass distributions.** To capture cell mass response to treatment with adequate statistical significance, we utilize the flow-through format of SMRs, enabling mass measurements of single cells[23–27]. An SMR sensor is composed of a suspended cantilever with an integrated U-shaped microfluidic channel[28] (Fig. 1c). As a cell passes through the integrated channel, the cantilever's mass is transiently altered, inducing a brief change in the resonant frequency proportional to the buoyant mass of the cell, referred to as "mass" throughout this paper (Methods). The fluidic control scheme implemented in the instrument (Supplementary Fig. 2),

together with the SMR chip, enables us to consistently measure samples of 5000 cells from a 50 µl volume in 10 min.

For measuring the treatment response from a patient specimen, we first isolate the cancer cells from the sample (Fig. 2a) and incubate the aliquoted cells with drugs or drug combinations (Methods). We then flow the cell populations through the mass sensor to capture their cell mass probability distribution functions, which we will refer to as mass distributions in this paper. As an example, Fig. 2a shows three distinct mass distributions—a reference distribution of vehicle-treated cells (gray) and two distributions of drug-treated cells (blue and purple). We compare the distributions of treated cells to that of the reference cells using Earth Mover's Distance (EMD), a measurement of statistical similarity, and quantify the difference as a "mass response" signal (Fig. 2b, Methods). We measure a larger mass response for diverging mass distributions (higher EMD value, blue versus gray) and a smaller mass response when mass distributions are similar (lower EMD value, purple versus gray). To achieve a malignancy-agnostic metric that can be used across various tumor specimen types, we normalize each single-cell mass measurement by the mean mass of the vehicle-treated cells in the sample, resulting in a unitless mass response signal that reports mass change (in percent) relative to control (Fig. 2b).

As with other population-based statistical tests, the accuracy of the mass response measurement relies on how well the sampled cell populations represent the true distribution in the tumor sample. To understand the impact of mass measurement parameters such as sensor noise and the number of cells measured, we ran simulations using the data shown in Fig. 2a (Supplementary Fig. 3a, b). Measuring at least 2500 cells to calculate a mass distribution limits the baseline noise in the mass response signal between identical samples to less than 1.5% and the standard deviation of the mass response to less than 1%, whereas when the measurement is based on a sample of 500 cells, these parameters are 3 and 1%, respectively.

Due to the inherent biological heterogeneity of patient specimens, the isolated single cells within a sample may exhibit

different levels of treatment response. Therefore, signal linearity is a critical attribute for accurately translating the treatment-induced mass change to a linear mass response that can be compared across samples, drugs, and treatment doses. As a demonstration, we simulate varying magnitudes of mass responses by sampling cells at different ratios from the treated and reference distributions shown in Fig. 2a. We show that the mass response signal is a linear function of the ratio of responding cells in the sample (Supplementary Fig. 3c).

**Interpreting mass response as cancer cell response to treatment.** A key goal of functional testing is to enable measurements to be performed in short timescales, ideally less than 48 hours, minimizing the impact of possible phenotypic drift and viability change of the primary cancer cells being tested. To ensure accurate and reliable treatment response results, we measure vehicle-treated cells twice—both before and after the treated cell populations—to account for any phenotypic drift over the course of mass measurement, which may take a few hours when testing several conditions. For most drugs presented here, dimethyl sulfoxide (DMSO) is used for dissolution, and therefore, DMSO (0.25%) treatment alone serves as the vehicle control. Mass measurements of cells treated with 0.25% DMSO are indistinguishable from untreated cells, suggesting a minimal effect of DMSO alone on cell mass (Supplementary Fig. 1). Figure 2c demonstrates the structure of the measurement approach. First, a population of vehicle-treated cells are measured to be used as a "reference" distribution. Then, cells that were exposed to treatment are measured. Multiple treatment "*conditions*" can be sequentially measured after the reference cells for testing a drug panel. Finally, a second replicate condition of vehicle-treated cells are measured as a "control". To quantify treatment-independent changes in the vehicle-treated cells throughout the measurement duration, we calculate the mass response between the vehicle-treated control and reference distributions (CTRL in Fig. 2d). To quantify cell response to treatment, we calculate the mass response between the drug-treated cells and vehicle-treated reference cells (TEST in Fig. 2d). Comparing the TEST and CTRL signals using bootstrapping[29] to confirm a signal magnitude difference larger than a "limit of decision" threshold yields a $p$-value for interpreting the treatment response outcome (Methods). For example, a distance measured between blue and gray dots in Fig. 2d greater than the limit of decision with a correspondingly low $p$-value would indicate that cells treated with the tested drug changed their mass relative to the control cells at a significant level. A high $p$-value rejecting the hypothesis would instead indicate no response to treatment (purple dot in Fig. 2d). In this paper, we define the three-sigma limit of decision[30] as 3% across all measurements, which corresponds to three times the standard deviation of the distance between 500 cells repeatedly sampled from a cell population (Supplementary Fig. 3a).

To test the robustness of our approach, we simulated cellular phenotypic drift in the form of mass loss as a function of time (Supplementary Fig. 3d). We tested varying rates of mass loss-per-time for cells to identify the limits of the measurement to correctly capture the response of the treated cells. Assuming a linear rate of phenotypic drift as a function of time, we find that for correctly resolving a mass response magnitude of 5% (relative to control), a phenotypic drift of vehicle-treated cells should be less than 10%. We have not observed phenotypic drift rates exceeding this number for any cell line or primary specimens reported here. Nonetheless, our approach enables us to identify significant phenotypic drift by monitoring the distance between the reference and control cells, as shown as the CTRL signal (Fig. 2d). If this distance is found to be higher than 10%, we conclude that the test is inconclusive due to high phenotypic drift.

**Image classification for identifying single cells of interest.** Despite the high efficacy of commercially available cell enrichment kits (Methods), processed primary tumor specimens often contain biological debris and cellular aggregates in addition to single cells of interest. Because mass measurements alone cannot distinguish between these different particle types, this additional material can interfere with the ability to detect drug-induced changes in mass distributions. To address this challenge, we implemented brightfield imaging inline with the mass sensor (Fig. 1) with real-time optical particle detection immediately downstream to trigger image capture. Each mass measurement is paired with its corresponding brightfield image and annotated using CNN-based image classification. This image classification occurs in two stages. First, a binary CNN classifier is used to identify which single-cell events to accept and non-single-cell events to reject, such as debris and cellular aggregates. Each accepted event is classified further as either an intact or permeable single cell and each rejected event is characterized as either an aggregate or debris using two additional binary CNN classifiers (Fig. 3a).

We trained the CNN models using manually curated images of each class collected for a range of cell lines and primary tumor specimens to ensure generalizability across various specimen formats (Methods). When applied to manually curated image sets, these models achieve cross-validated precision and recall values exceeding 97% for each image class (Fig. 3a).

In this training set, images with small particulate matter or fibrous material are classified as debris, and images with clearly segmented clusters of cells are classified as aggregates. Intact and permeable cell discrimination is based primarily on the reduced contrast seen in cells that have presumably lost their membrane integrity. This loss of contrast observed by brightfield imaging is consistent with viability data collected in parallel with flow cytometry assessment of DAPI, a DNA-intercalating dye that is more accessible to the nuclei of non-viable cells that have lost a functioning cell membrane (Supplementary Fig. 4).

The utility of image-annotated single-cell mass measurements can be seen when comparing the mass distributions of each particle class for cell lines with different baseline mass characteristics (Fig. 3b). Human lung cancer cells (PC9) and human multiple myeloma cells (MM1S) have significantly different underlying mass distributions. Given this variation, relying on mass measurements alone to identify single cells versus debris or aggregate events based on universal gating would not be feasible across various cell types. In contrast, image annotation and classification offer a broadly applicable means of identifying particles of interest across various samples, as can be seen with the consistent mass trends across particle classes observed in these two cell lines, with cell aggregates having the largest mass followed by intact cells, permeable cells, and debris. This flexibility allows for the identification of single cells for further analysis regardless of the underlying structure of a given specimen's mass distribution. This improved ability to characterize single-cell mass distributions is a key requirement for robustly identifying treatment responses. To quantify the benefit of linked imaging, we compared the sampling error between random subsets of cells drawn from either all mass measurements collected in a condition or only the mass measurements annotated as accepted by image classification (Fig. 3c). Across 3222 different datasets collected for a range of primary cells and cell lines, we found that image curation significantly improved this sampling variability, with an average decrease in sampling

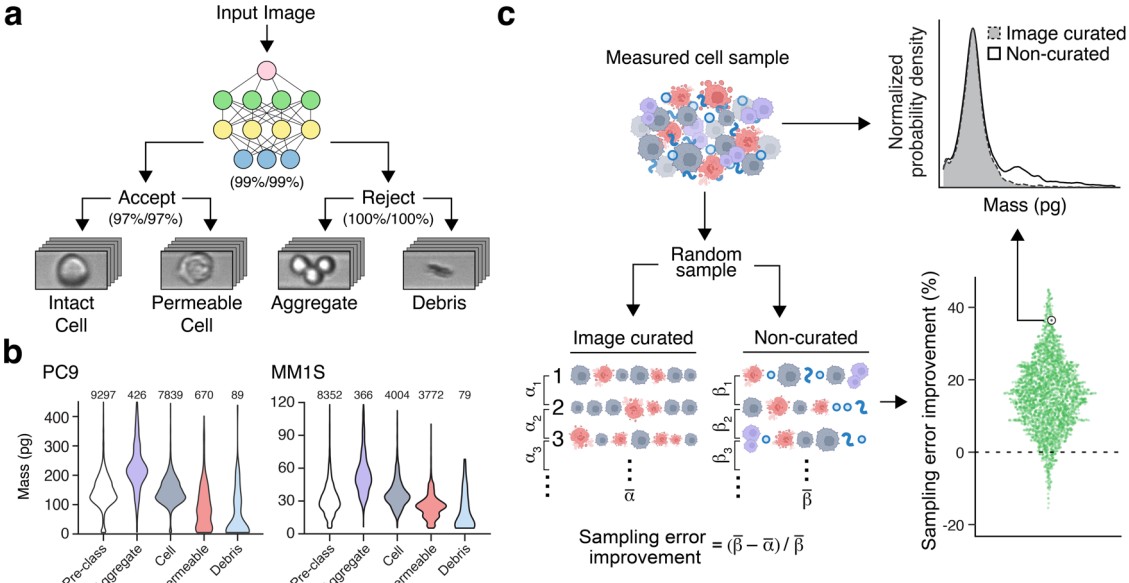

**Fig. 3 Image annotation of mass data improves measurement fidelity. a** Schematic representation of the multi-step image classification approach implemented in the workflow depicted in Fig. 1. An input image is first classified as accepted or rejected with a binary CNN classifier, and subsequently, all accepted particles are classified as either an intact or permeable cell and all rejected particles are classified as either an aggregate or debris with two additional binary CNN classifiers. The inset images are representative of each class. The parenthetical percentages listed at each binary decision node indicate the cross-validated model performances, listing the precision and recall values, respectively (Methods). **b** Violin plots showing the mass distributions for a full population of cells without image classification (white) as well particles within the population that were classified as aggregates (purple), intact cells (gray), permeable cells (pink), or debris (blue) for a human lung cancer cell line model (PC9) and a human multiple myeloma cell line (MM1S). **c** Single-cell image classification reduces the noise in mass response signal by reducing the sampling error. As a demonstration, we randomly sample 1000 cells 100 times from a measured condition with and without image curation and calculate the mean mass response within these two sets representing the sampling error with and without image curation. When this is repeated for all the measurements presented in this paper, including 3222 measured conditions across 13 different instruments, we find that image curation reduces the sampling error by 16.2% on average. An example pair of mass distributions with (gray) and without (white) image curation is shown, indicating the presence of cell aggregate and debris populations in the original measurement set, which are subsequently removed by the image curation process.

error of 16.2% when compared with non-curated measurements from the same condition. Interestingly, "Aggregate" events appeared to account for more sampling error than "Debris" events when considering each class individually (Supplementary Fig. 4b, c). These results demonstrate the ability of linked imaging to improve the reliability of sampling an underlying mass distribution, particularly in the context of highly heterogenous specimens where only a limited subset of the measurements are single cells.

**Cellular mechanisms of mass response**. When considering changes to cell mass that may occur in response to treatment, we define three potential categories related to a drug's mechanism of action (MOA): (1) changes due to cell cycle arrest, (2) changes due to disruption of metabolic processes, and (3) changes due to failure of the cell's structural integrity (Fig. 4a). Using a basic set of assumptions about the nature of each type of drug response mechanism, we can create simple models that demonstrate the expected changes in mass across a population of single cells. In the case of cell cycle arrest, we expect that the mass distribution gradually consolidates around the average newborn-cell mass for G0/G1 arrest or around the average cell mass prior to the division for G2/M arrest (Fig. 4b). Prior work has shown that disruption of metabolic pathways can manifest as changes to single-cell mass[31]. These effects can skew towards anabolic or catabolic processes, resulting in larger or smaller cells, respectively, as shown schematically in Fig. 4b. Disruption of cell structural integrity is expected to lead to the largest changes in mass across a population, as this category is consistent with apoptosis and/or

necrosis of cells, and mass loss is likely driven by dramatic physical changes to cells such as the loss of membrane integrity/cytoplasm, membrane blebbing, cellular fragmentation, and other processes. Here, we assume that large quantities of mass loss will cause cells to shift from their initial vehicle-treated distribution towards a minimally overlapping secondary distribution (Fig. 4b).

Drugs with the MOAs described here are reasonably common, as are homogenous cell lines that respond to these drugs, allowing us to test these hypothetical models. To test the mass response outcome of G0/G1 arrest, we exposed the human lung cancer H1666 cell line to 10 uM trametinib, a MEK inhibitor, for 17 h, which arrests most cells in early G1 (Supplementary Fig. 5a)[32]. Consistent with expectation, we saw the cell mass distribution shift downward as compared to the control (Fig. 4c, d). In contrast, by treating MDA-MB-361 cells for 24 h with 10 nM docetaxel, a microtubule inhibitor which prevents cell division, we observe a significant upward shift in mass (Fig. 4c, d and Supplementary Fig. 5b). These two cell cycle arrest phenotypes are central to the activity of many drugs, both targeted inhibitors and chemotherapies, and mass response resolves these phenotypes robustly across many examples (Supplementary Fig. 6). To produce metabolic skew data, we treated cells with cycloheximide, a ribosomal inhibitor, or carfilzomib, a proteasome inhibitor, to skew metabolism towards catabolism or anabolism, respectively. In L1210 cells treated with 400 nM cycloheximide for 24 h, we observe a decrease in the average cell mass in the population, consistent with inhibition of protein biogenesis[33]. If we look at U266 cells treated with 50 nM of the proteasome inhibitor Carfilzomib for 6 h, we instead observe a subtle increase in the average mass of cells, consistent with excess protein accumulation

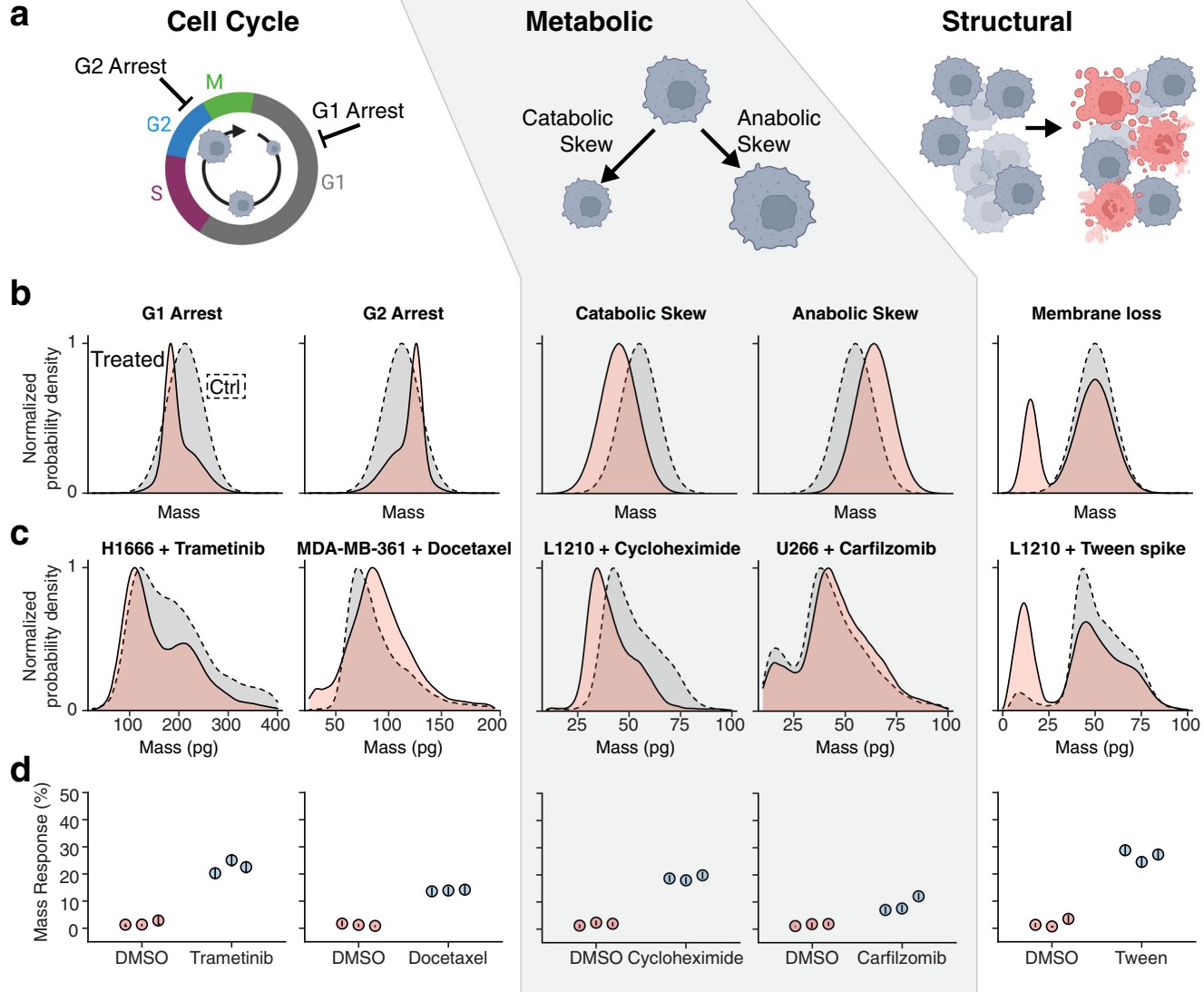

**Fig. 4 Mass responses to drugs with various mechanisms of action.** Therapies with different mechanisms of action (MOA) induce different characteristic mass response signatures. As depicted in (**a**) these MOA signatures capture drug effects that can be loosely classified into three general categories, including cell cycle arrest, metabolic disruption, or loss of structural integrity. **b** Schematic demonstrations of the changes in mass distributions between a control population (gray fill, dotted line) and a drug-treated population (red fill, solid line) caused by G1 arrest, G2 arrest, catabolic skew, anabolic skew, or cell membrane loss. **c** Example experimental mass distribution changes in response to drug treatment corresponding to each schematic result presented directly above in (**b**). **d** Mass response measurements collected with three different instruments (individual points with error bars corresponding to the 95% confidence interval of EMD for each system defined by 5000 random samples of 2500 cells), corresponding to the experimental data presented directly above in (**c**).

prior to downstream cytotoxicity (Fig. 4c, d)[34]. Finally, as an example of complete structural disruption, we used L1210 cells treated with 0.5% Tween 20 detergent for 10 min, which permeabilizes the cell membrane and spiked in at 40% to an otherwise healthy cell population. Here, we observe a decrease in the primary mass peak representing live cells, and an increase in the smaller peak, which represents permeabilized cells (Fig. 4c, d). These same changes to mass distributions can be observed for cells following cell death induced by clinically relevant drugs (Supplementary Fig. 7).

The ability of mass distributions to discern these different response profiles makes it well-suited for detecting drug response in heterogeneous primary samples. The dynamic nature of these mass change mechanisms, combined with heterogeneity in the timing of cellular response and potential phenotypic drift ex vivo, means that the presentation of these mechanisms is not necessarily uniform across a population of cells. For example, even in homogenous cell lines, such as PC9 cells treated with doxorubicin, different doses of the drug at a single timepoint show how mass response signal can be manifested from both cell cycle arrest and structural disruption, either alone or simultaneously (Supplementary Fig. 8).

**Mass response modulates with dose, time, and cellular fraction.** In addition to compatibility with various drug mechanisms, when considering the role of mass response measurement in a clinical pipeline, it is also important to demonstrate compatibility with heterogeneous tumor cell specimens that have a limited time window of phenotypic stability ex vivo during which drug sensitivity can be assessed. It is, therefore, important to assess the effects of drug concentration and time, as well as underlying response heterogeneity on mass response readouts.

The canonical approach used to characterize drug sensitivity as a function of dose and time is the viability-based dose-response curve or $IC_{50}$ curve. As such, a range of viability markers and techniques (i.e., MTT, ATP, flow cytometry) have shown potential as functional biomarkers for cancer care, but the impact of these approaches in the context of primary tumor samples has often been limited by the number of cells required and the time necessary to conduct these assays[1]. However, as an established standard, $IC_{50}$ curves provide a useful comparator and model for understanding the variables that affect cell drug response. $IC_{50}$ measurements sweep dose space to define the dose inflection point above which a majority of cells die in response to a drug. The optimal timepoint for assessing viability-based dose response is typically dictated by the drug mechanism and cell line being studied. For fast-acting drugs, a 24 h timepoint is often sufficient to accurately define cellular sensitivity; however, for slow-acting drugs (e.g., drugs functioning through cell cycle arrest), a timepoint of 72 h or longer may be required.

To evaluate how these dose concentration and timing parameters affect cell mass response, we modulated these variables independently in cell lines to understand their effect. MM1S cells treated with a range of concentrations of carfilzomib for a fixed amount of time (15 h), showed a dose-dependent mass response similar to the $IC_{50}$ curves collected for the same cell line (Fig. 5a). We also noted that mass response changes over time in response to a fixed concentration of drug (Fig. 5b). For fast-acting drugs, which rapidly induce cytotoxicity, mass response magnitude demonstrates a dose dependence in line with viability loss measurements collected at similar timepoints (Fig. 5c and Supplementary Fig. 9a, b). However, as a result of being able to detect signals prior to cell death, mass response can detect the effects of slow-acting drugs well in advance of a 50% viability loss required to define an accurate $IC_{50}$ signal (Fig. 5d and Supplementary Fig. 9c, d). For example, in the case of PC9 cells treated with paclitaxel, 24-h mass measurements accurately define a dose-response inflection point revealing effective concentrations of the drug, despite minimal changes in viability observed at all concentrations tested and $IC_{50}$ measurements requiring 72 h for a more accurate readout (Fig. 5d). When this comparison is made across 60 different drugs tested across 12 different cell lines, the value of this rapidly manifesting mass response signal is made clear (Fig. 5e and Supplementary Data 1). For many drugs, whether targeted kinase inhibitors or chemotherapies, a 24-h $IC_{50}$ value is comparable to measured mass response with mass-based signal developing at the same or only slightly lower doses of the drug than observed by viability measurements. However, for drugs which work through slower-acting mechanisms such as cell cycle arrest, 24-h mass response measurements still define effective drug concentrations, whereas 24-h $IC_{50}$ measurements provide little perspective. Instead, 72-h or longer $IC_{50}$ timepoints must be taken to define cellular sensitivity to such drugs (Fig. 5e). This ability to rapidly detect drug-induced changes in cellular phenotype is particularly beneficial in the context of primary tissue measurements where longer-term drug incubations are infeasible due to phenotypic drift and viability loss ex vivo.

While homogenous cell lines provide a good context for probing the fundamental characteristics of mass response measurements, they are not good proxies for the heterogeneity seen in primary tumor specimens. For this reason, it is important to assess the impact of heterogeneity on mass response measurements. This variable can be probed explicitly by mixing drug- and vehicle-treated fractions from the same cell line, demonstrating that at a given timepoint, the mass response increases proportionally to the fraction of cells responding (Fig. 6a). A more complex model is needed to emulate the heterogeneous size distribution and drug sensitivity in a primary sample. To test fractional sensitivity with this heterogeneity in mind, we used a mixture of three cell lines, each with a unique sensitivity to an individual drug (Fig. 6b). When we observe response using 1-drug versus, 2- or 3-drug combos, we see an additive shift in mass response that is roughly the sum of responses for each drug as a monotherapy (Fig. 6b).

These results demonstrate a high degree of concordance between mass response measurements and other existing drug response assays and show that mass response can accurately characterize the effects of time, dose, and sample heterogeneity. The higher information content offered by single-cell mass response measurements and their unique ability to resolve cellular sensitivity at earlier timepoints upstream of viability loss offer clear advantages in characterizing primary tumor cells where longer-term maintenance of cells ex vivo is not practical.

**Demonstrating the feasibility of mass response measurements for various specimen formats.** Sample composition and collection feasibility vary significantly across different clinical specimen formats and can affect the ease of cell isolation and measurement. A functional testing pipeline must therefore be compatible with specimens collected from a variety of tumor cell compartments in order to maintain broad applicability across malignancies.

Previous work has demonstrated the feasibility of using mass response measurements to characterize drug efficacy for hematological malignancy sample formats such as blood and bone marrow[21]. While providing an encouraging proof-of-concept that such biophysical readouts can accurately predict patient responses to therapy, the technical complexity of processing solid tumor specimens led us to test whether our image-annotated mass measurement workflow could maintain the ability to characterize drug sensitivity while also offering the speed, robustness, and technical reproducibility required of a clinical testing pipeline (Supplementary Fig. 2b).

To first demonstrate the compatibility of this workflow with hematologic tumor specimens, we present measurements of a peripheral blood sample from a patient with plasma cell leukemia (PCL), and a bone marrow aspirate from a patient with multiple myeloma (MM) (Fig. 7a, b). In both cases, cellular mass responses were observable for a range of therapies. Tumor cells isolated from the PCL sample with a prior demonstration of the t(11;14) translocation showed a dose-dependent response to venetoclax as a monotherapy and when in combination with bortezomib and selinexor. However, these cells did not show a significant mass response to selinexor or bortezomib alone, suggesting that the response was driven primarily by venetoclax. This patient had previously been treated with venetoclax-based therapy and had a good response lasting five months, as indicated by eradication of the t(11;14) clone in subsequent diagnostic bone marrow biopsies. However, treatment was discontinued after five months due to adverse effects (cytopenias). The bone marrow aspirate sample from a relapsed/refractory multiple myeloma patient with known extramedullary involvement demonstrated a dose-dependent response to the combination of selinexor, carfilzomib, and dexamethasone, as well as the DCEP therapy combination (dexamethasone, cyclophosphamide, etoposide, and cisplatin). When dosed as monotherapies, most of the mass response observed with the combination therapy was recapitulated by dosing with the cyclophosphamide analog mafosfamide—a spontaneously hydrolyzing compound that produces the same two components as metabolized cyclophosphamide[35]. As with prior predictive measurements in multiple myeloma, DCEP mass response measurements were consistent with the patient's decrease in serum monoclonal protein and response to treatment with salvage combination chemotherapy.

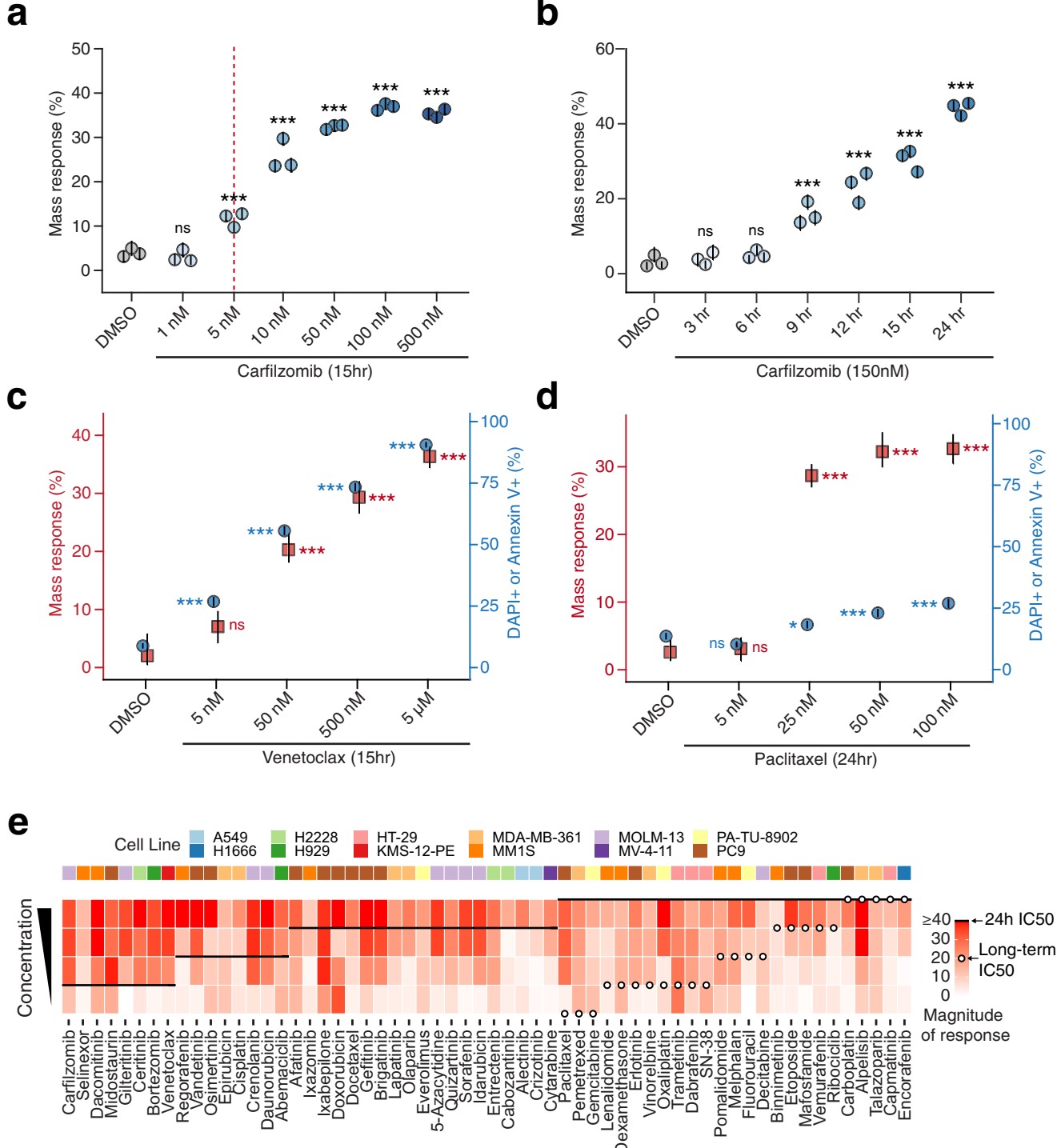

**Fig. 5 Mass response changes with dose and time, and can precede cell viability loss for slow-acting drugs.** Mass response signal varies with changes to dose or timing of measurement, similar to cell viability measurements. Mass response of MM1S cells treated with carfilzomib shown when **a** varying dose at a 15 h timepoint or **b** varying timepoint with a 150 nM dose. Multiple points indicate independent instruments, overlaid lines indicate a 95% confidence interval for mass response defined by 5000 random samples of 2500 cells. **c** MM1S cells treated with the fast-acting drug venetoclax for 15 h, and **d** PC9 cells treated with the slow-acting drug paclitaxel for 24 h, comparing dose response as measured by mass (red axis, red squares show representative mass response measurements, overlaid lines indicate 95% confidence interval), or flow cytometry-based assessment of viability using DAPI and Annexin V (blue axis, blue circles, overlaid lines indicate 95% confidence interval) (Methods). **e** Heatmap showing a mass response of cell lines (top legend) to 60 different drugs, comparing the magnitude of response (red gradient; as determined by the difference between control and the test conditions) for rank-ordered doses to 24 h IC$_{50}$ (black lines) or long-term IC$_{50}$ values (open circles). Long-term IC$_{50}$ values are only provided for drugs where 24 h IC$_{50}$ is infinite or greater than the highest useable concentration for testing. Drugs are ordered in the x-axis with respect to their 24 h IC$_{50}$ values. Observable mass response at doses below short and long-term IC$_{50}$ measurements for a given cell/drug combination indicate that mass changes either correspond to or precede viability loss for all drugs presented. *** indicates $p < 0.001$, ns indicates $p > 0.05$.

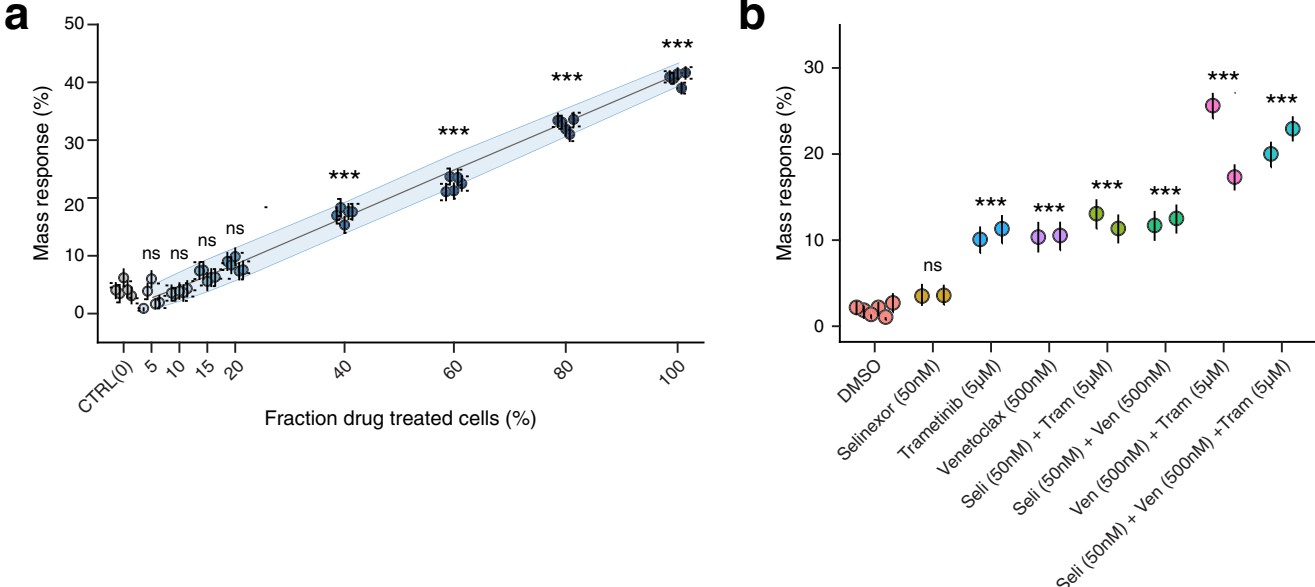

**Fig. 6 Mass response in the context of heterogeneous drug sensitivity. a** Mass response measurements taken on MM1S cells, where vehicle control and carfilzomib treated cells were mixed after treatment to create cell populations with defined fractions of responding cells. **b** A 1:1:1 combination of MM1S, H929, and KMS-12-PE cells treated for 15 h with either single-, double-, or triple-agent therapies, demonstrating the additive signal seen as the sensitive fraction of cells increases in response to combination treatment. Multiple points indicate independent instruments, overlaid lines indicate a 95% confidence interval for mass response defined by 5000 random samples of 2500 cells. *** indicates $p < 0.001$, ns indicates $p > 0.05$.

For patients with relapsed or metastatic solid tumor malignancies, clinical assessment often requires the collection of solid tissue samples rather than blood or bone marrow specimens. Collection of these specimens by means of surgical resections or sometimes even by core biopsies are infeasible given the size and anatomical location of metastatic lesions, and the desire to avoid unnecessary invasive procedures. Fine-needle aspiration (FNA), which utilizes lower profile needles as compared with core biopsies, offers a minimally-invasive alternative to sample collection and reduces the risk of bleeding and injury[36]. To ensure maximal clinical utility, we sought to determine the feasibility of performing mass response measurements using these low-input FNA specimens, which often yield only tens of thousands of single cells for downstream analysis.

We collected FNA specimens from three different anatomical locations: a lung mass in a patient with non-small cell lung cancer (NSCLC), a neck mass in a patient with melanoma, and a soft tissue lytic bone mass in a patient with breast cancer (Fig. 7c). Total cell yields were 115-, 25-, and 120-thousand tumor cells for the lung, bone, and neck masses, respectively. These specimens demonstrated a range of cell mass drug responses, with the lung mass showing a significant mass response to paclitaxel and gemcitabine, the neck mass showing a significant mass response to a combination of dabrafenib and trametinib, and the soft tissue bone mass showing no significant response to docetaxel or doxorubicin. Interestingly, the patient with NSCLC was subsequently treated with a combination of carboplatin and paclitaxel and demonstrated a marked clinical response, consistent with the mass response to paclitaxel noted for this specimen. These measurements demonstrate the feasibility of performing the end-to-end workflow with low-input tissue formats and are an indication that the pipeline is compatible with performing drug response testing within the constraints of current clinical management strategies for patients with advanced solid tumor malignancies.

In addition to disseminated metastatic lesions, many patients with advanced cancer accumulate malignant fluids in the form of pleural effusions or abdominal ascites, which cause significant discomfort and must be drained for diagnostic and therapeutic reasons to manage symptoms[37]. Because these malignant fluids contain tumor cells, they offer another potential specimen format for minimally-invasive drug response testing. After standard tumor cell enrichment protocols (Methods), these samples often yield a significant number of cells for measurement. For example, in a patient with advanced non-small cell lung cancer, a 150 ml sample of malignant pleural effusion yielded nearly 170 million tumor cells, more than enough to perform mass response testing for a panel of drugs (Fig. 7d). This patient had been undergoing treatment with capmatinib due to a confirmed MET exon 14 skipping mutation but had not been responding to this therapy at the time of the effusion collection. Mass response measurements collected on the tumor cells isolated from the effusion sample were consistent with this clinical outcome, revealing no significant mass response to capmatinib across doses ranging over multiple orders of magnitude. However, these cells were not generally unresponsive to all treatments, showing significant and dose-dependent mass responses of varying magnitudes to therapies including paclitaxel, docetaxel, and cisplatin (Fig. 7e and Supplementary Fig. 10a). Consistent with cell line measurements of slower-acting taxane drugs—including paclitaxel and docetaxel—the mass responses detected for these drugs were not observable with flow cytometry-based viability measurements collected for this same specimen (Supplementary Fig. 10b). These results demonstrate the feasibility of collecting mass response measurements with malignant fluid specimens and provide an example of the drug response heterogeneity that can be revealed by measuring primary tumor cells directly. Additionally, they demonstrate the potential of this new approach to complement existing genomic biomarkers, which, as in the case of this patient, do not always identify an efficacious therapy.

## Discussion

The workflow presented here builds on recent efforts in functional precision medicine for cancer[10–17] and offers key

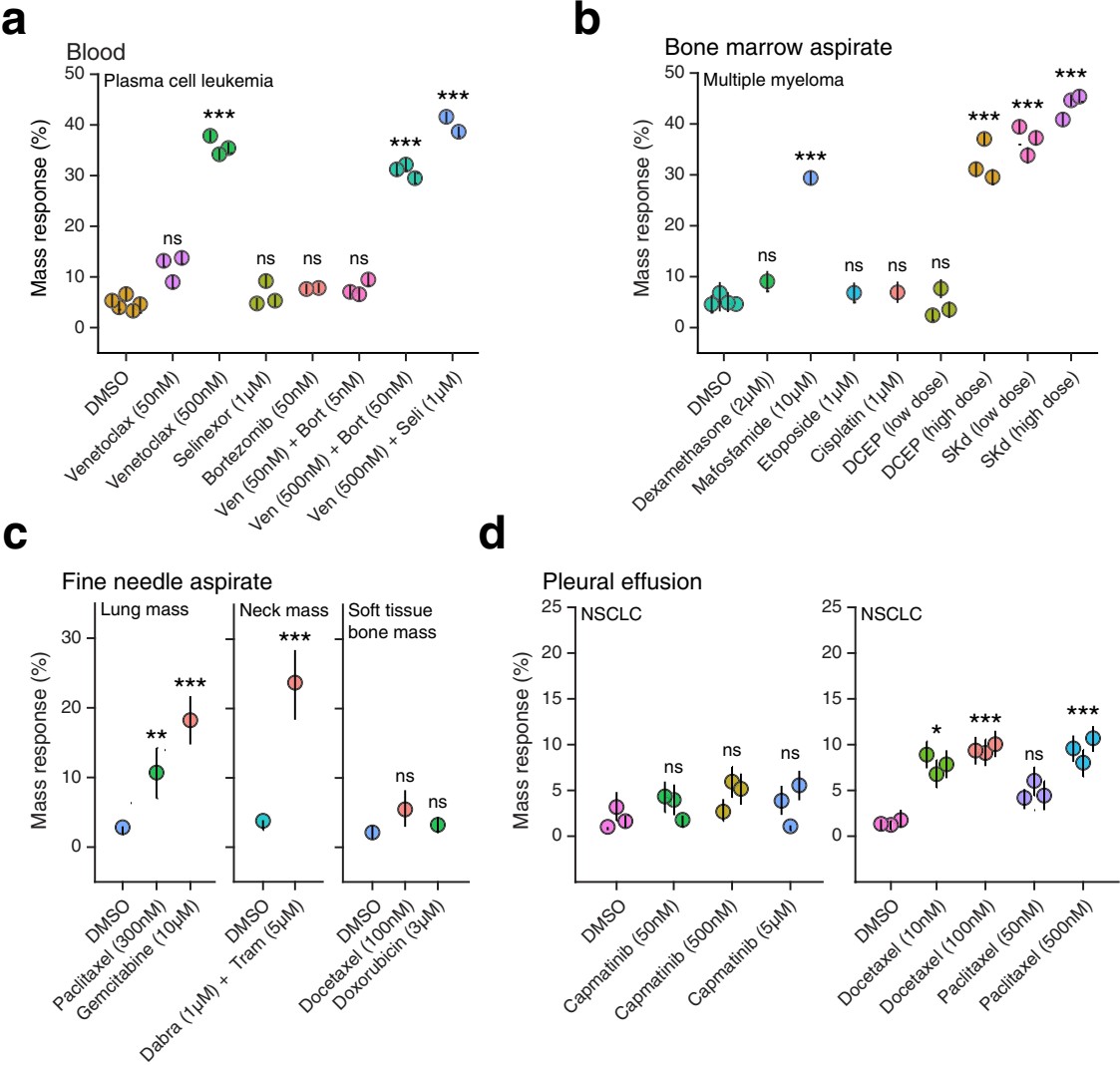

**Fig. 7 Mass response assessment of drug sensitivity for various cancer specimen formats.** Mass response measurements were collected for **a** plasma cell leukemia (PCL) cells isolated from a peripheral blood sample and treated for 15 h with the drugs listed, **b** multiple myeloma cells isolated from a bone marrow aspirate sample and treated for 15 h with the drugs listed, **c** cells isolated from fine-needle aspirate (FNA) samples collected from a lung mass, neck mass, and soft tissue bone mass in three different patients with non-small cell lung cancer, melanoma, and breast cancer, respectively, and treated for 20 h with the drugs listed, and **d** non-small cell lung cancer (NSCLC) cells isolated from a pleural effusion sample and treated for 20 h with the drugs listed. For all plots, each point represents the mass response measured in an individual instrument, with error bars corresponding to the 95% confidence interval for mass response defined by 5000 random samples of 2500 cells. *** indicates $p < 0.001$, ** indicates $p < 0.01$, * indicates $p < 0.05$, ns indicates $p > 0.05$.

additional translational developments towards broadly applicable rapid therapy guidance for clinical cancer care. We have demonstrated the feasibility of collecting drug response measurements across cyto-dilute and minimally-invasive specimen formats used to sample solid tumors such as FNA and malignant fluids, in addition to those commonly used in the study of hematologic malignancies, including blood and bone marrow. The technical improvements described here, including image classification for measurement curation and the statistical approach, assist in identifying treatment responses using a limited number of cells from a heterogeneous and labile primary sample. The reduction of background noise in primary samples enabled by these advancements also serves to shorten the turnaround time of the test to 2 days by limiting the duration of drug exposure necessary to observe the response.

In addition to demonstrating broad utility across various malignancies and specimen formats, we have also shown that this workflow can be used to identify mass responses for a wide range of therapies. Despite unique mechanisms of action and differing effects on cellular biophysical properties, single-cell mass measurements were able to identify dose-dependent responses in cell lines for all the major drug classes tested here. Furthermore, with few exceptions, these mass responses were observable within 24 h, often before a significant signal could be detected with alternative viability-based measurements. For exceptionally slow-acting drugs, such as the anti-metabolite 5-fluorouracil, although the mass response was not observable until roughly 48 h after treatment, this preceded any observable change in cell viability as assessed by flow cytometry (Supplementary Fig. 9c). Together, these results suggest that mass response measurements offer a degree of generalizability on par with existing gold-standard, viability-based drug response assays while enabling a faster response readout in many contexts—an essential set of features when characterizing heterogeneous clinical specimens that may be undergoing rapid phenotypic drift ex vivo[38,39].

As with other drug response measurement approaches, there are certain limitations to using mass response for rapid therapy guidance. For instance, due to the tumor cell isolation and drug dosing approaches used in this workflow, it is optimized for characterizing drugs with tumor cell-intrinsic mechanisms of action. Therapies that act primarily through non-cell intrinsic means, such as anti-angiogenic drugs that require physiological remodeling (e.g., bevacizumab) or endocrine therapies that indirectly target hormone receptor-positive cancers (e.g., aromatase inhibitors), are currently incompatible with this workflow. However, certain therapy types, including immuno-oncology drugs such as checkpoint inhibitors, may require less significant pipeline alteration to implement effective drug response measurements. Despite these therapies acting through non-tumor cell-intrinsic mechanisms, previous work has demonstrated that single-cell mass also provides a readout of the functional state of immune cells[21,22]. The workflow presented here is amenable to the direct measurement of these immune cell-intrinsic drug effects.

The single-cell nature of this mass measurement pipeline offers additional technical opportunities moving forward. For example, linked imaging was used here for particle classification and mass measurement data quality improvement, but the optical access offered by this platform is compatible with additional linked single-cell readout approaches, including brightfield image analysis for feature extraction, or fluorescence detection for immunophenotyping. In combination with fluidic handling developments, these optical improvements may lead to lower cell number requirements for future versions of the assay. As a non-destructive method, these single-cell mass measurements can also be collected upstream of linked multi-omic single-cell molecular readouts, as has been demonstrated previously with paired scRNA-seq[22]. Similarly, mass response measurements collected for a primary tumor specimen can be used to complement genomic results collected for the same patient, either simultaneously to improve predictive accuracy, or downstream to select the optimal inhibitor for a given genomic alteration. In either case, single-cell mass measurements offer a functional assessment of therapeutic response to further parse the underlying biological determinants of drug sensitivity, offering significant potential benefits for both clinical decision-making and cancer drug development.

As a functional test targeting a wide breadth of malignancies and drug mechanisms, this approach presents unique challenges. For example, the scope of the drug response testing possible for a given primary specimen is largely dependent on the total amount of tumor cells available after processing. As can be seen in Fig. 7 and Supplementary Fig. 11, the number of conditions tested can therefore range from one to over 25. This cell yield is highly variable across specimen formats and individual samples and is an important consideration when designing drug panels to test for various sample types. Tumor cells isolated from primary specimens may also have highly variable purity and viability across different samples, another key consideration for implementing this workflow clinically (Supplementary Note 1).

Interpretation of the mass-response data collected for these primary specimens is also a key developmental focus of this approach as it moves into the clinic. While here we have focused on a simple binary assessment of response and non-response based on a statistical distance comparison, future work will focus on the value of considering the magnitude of response or other single-cell mass distribution features in capturing the depth or durability of clinical response. Context-appropriate validation datasets are required to ensure that the test correctly identifies clinically relevant response and non-response. These drug and malignancy-specific limit of decision specifications are the focus

of multiple ongoing prospective studies in acute myeloid leukemia, multiple myeloma, and solid tumors, including breast and lung cancer (NCT04985357 and NCT03777410). The work shown here has enabled the incorporation of this test into a robust and scalable CLIA-certified workflow for these and future clinical validation studies.

This demonstration of clinical feasibility, in addition to the fundamental properties of mass response measurements—drug and malignancy-agnostic utility, compatibility with small sample size, and rapid turnaround time—represents a significant step towards the broad clinical impact of a functional test on patient care.

## Methods

**Single-cell mass measurement**. We use the SMR technology to conduct single-cell measurements in a similar way to what was previously presented[23,28], at flow rates exceeding 80 nl/s and with mass precision of less than 0.5 pg. Different from the previous studies, the measurement setup used for this study is an integrated instrument that controls the SMR sensor and associated fluidics. Thirteen identical SMR instruments were utilized to collect the data presented in this paper. Each instrument is equipped with a pressure- and flow-based fluidic control setup, a temperature controller for keeping the SMR chip stable at room temperature, an FPGA controller along with custom readout and drive electronics for controlling the SMR sensor and record data similar to that described in Olcum et al.[40], an imaging unit to capture images of single cells as they flow through the SMR and a computer to run the control and analysis software. The SMR and the sub-units in the prototype are controlled by custom software developed in the LabView environment. The software automatically guides the technician through the steps of calibration, fluidic priming, cell measurement, and cleaning steps sequentially. The cleaning step is performed between each measured drug condition to prevent contamination across measurements. The settings and parameters of measurements for the instruments are identical across all the SMR instruments and are pulled by the control software from a local server. Furthermore, the control software is equipped with automatic routines such as flow-kick-back for clog prevention, automatic sample loading for minimizing waste, and cleaning routines to increase speed and reliability.

**Calculation of mass response signal**. In this work, we use Earth Mover's Distance (EMD), also known as the first Wasserstein Distance, to quantify the difference of mass distributions as "mass response". EMD possesses properties of an ideal metric to quantify differences in cellular distributions[41] such as linearity, translation invariance, symmetry, and is robust to small differences of distributions due to instrument drift, measurement noise, or other sources of variability[42]. As suggested by Orlova et al.[42], EMD meets all the requirements given above and is computationally efficient for executing one-dimensional data like the single-cell mass measurements[43].

To define the mass response signal as a normalized, dimensionless metric to compare mass distributions, we introduce the following notation (Eq. 1):

$$\text{Mass Response of } X := \text{EMD}(X, Z)/\sum_i^N Z_i = \sum_i^N |X_i - Z_i|/\sum_i^N Z_i, \quad (1)$$

where $X$ and $Z$ are sorted single-cell mass measurements of drug- and vehicle-treated reference cells, respectively, and $N$ is the number of cells in each distribution[42]. (Mass response is also calculated for instances where X and Z have different sizes. In the above equation, we assume N for both for simplicity.)

We calculate the 90% confidence interval of the mass response signal using non-parametric bootstrap methods as we cannot expect Gaussian (or other known) distributions for the cellular mass. Specifically, the $BC_a$ (bias-corrected and accelerated) method is used to estimate confidence intervals with accurate coverage probabilities over the entire range of mass response signal[44]. For all the reported mass responses in this paper, $N = 2500$ cells and the number of bootstrap replicates for the mass response confidence interval is $R = 5000$ unless otherwise explicitly noted.

**Statistics and reproducibility—determination of test outcome based on limit of decision**. The proposed structure for measuring mass response (Fig. 2c, d) enables us to test the biological significance of the measured mass response signal as opposed to directly comparing single-cell mass measurements using a statistical test. Since the number of cells measured ($N$) to represent each cell population is on the order of thousands, small deviations in the mass distributions due to sampling error, instrument noise, or phenotypic drift can turn out to be statistically significant. Such deviations, however, can be statistically significant without representing a biologically meaningful signal[45]. To circumvent this problem, we define the test statistic, $\theta$, to be the difference between two mass response signals, i.e., CTRL and TEST, as shown in Fig. 2d, and check if $\theta$ is larger than the limit of

decision threshold. Hence (Eq. 2):

$$\theta(X, Y, Z) \equiv \text{Mass Response}(X) - \text{Mass Response}(Y) = (\sum_i^N |X_i - Z_i| - \sum_i^N |Y_i - Z_i|)/\sum_i^N Z_i, \tag{2}$$

where $Y$ is the sorted mass measurements of the vehicle-treated control cells and $X$ and $Z$ are defined as above. The second term of the equation is the distance between reference and control cells measured at different points in time (Fig. 2c). The test statistic is thus the distance of the drug-treated cells to the vehicle-treated reference cells relative to the "phenotypic drift" (if any) in the control cells over the duration of the test. Even in the absence of drift, the second term captures the naturally occurring noise inherent in estimating the distance between two finite samples from the same distribution.

To test for cancer cell sensitivity to treatment, we compare the signal, $\theta$, to a threshold, $\theta_0$—the biologically meaningful "limit of decision"—appropriate for the class of drug being evaluated. It is challenging, in general, to do non-parametric hypothesis testing for non-zero nulls. As stated by Chernick, "Particularly important in hypothesis testing is the use of asymptotically pivotal statistics and centering the distribution under the null hypothesis"[46]. Consistent with the statement, we use the bootstrap-$t$ method as outlined here. Using the derivation of Davison et al.[29], we introduce a pivot, i.e., a combination of data and parameters whose distribution is independent of an underlying model (Eq. 3),

$$T = (\hat{\theta} - \theta)/S \tag{3}$$

where $\hat{\theta}$ is the observed signal, $\theta$ is the (unobservable) true signal, and $S$ is the standard error of $\hat{\theta}$. In this formulation, $T$ is asymptotically Gaussian. In the work presented here, $\hat{\theta}$ is measured from $N = 2500$ cells each of reference, control, and test populations. $S$ is estimated using $r = 19$ bootstrap replicants of $\hat{\theta}$.

To test the null hypothesis $H_0 : \theta \le \theta_0$, we substitute $\theta_0$ for $\theta$ and the observed value of the pivot under the null becomes (Eq. 4):

$$t_{obs} = (\hat{\theta} - \theta_0)/S \tag{4}$$

To do the hypothesis test, we compare $t_{obs}$ to a bootstrap simulation of $T$ (Eq. 5):

$$T^* = (\hat{\theta}^* - \hat{\theta})/S^* \tag{5}$$

where $S^*$ is the estimated error of the bootstrap replicant, $\hat{\theta}^*$. Since $S^*$ is, again, found by bootstrap, we have an "embedded bootstrap" method. We used $R = 999$ iterations of $\hat{\theta}^*$ and, for each of those, $r = 19$ iterations to estimate $S^*$.

The $p$ value, i.e., probability of achieving the observed result given the null hypothesis is true, is then (Eq. 6):

$$p\,\text{value} = \Pr(T \ge t_{obs}) \cong \frac{1 + \#\{T^* \ge t_{obs}\}}{1 + R} \tag{6}$$

where $\Pr(x)$ is the probability that $x$ is true and $\#[x^*]$ is the count of bootstrap variants for which $x$ is true. For the chosen number of iterations, $R$, the minimum achievable $p$ value is 0.001. We compare the resultant $p$ value to the customary significance level of 5%, i.e., $p$ value < 0.05, to determine if the null hypothesis can be rejected.

**Image classification**. Images collected of various cell types on the SMR instruments were manually curated to generate training sets of at least 10,000 images for each image class (intact cell, permeable cell, debris, and aggregate). Three different binary classification CNN models were then generated using the Keras library (2.3.1) in Python (3.7.7). These include (1) an accept/reject model that utilized both intact and permeable cell images as "accepted" events and both debris and aggregate images as "rejected" events during training, (2) an intact/permeable model, and (3) an aggregate/debris model. For training, we used image augmentation to improve the robustness of the final models. This augmentation included random vertical and horizontal image flipping, brightness adjustment, and rotation. Each model was trained using 90% of the images from each curated training set, with the remaining 10% used for cross-validation to test model performance. The precision and recall of this cross-validation are reported in Fig. 3. Images were classified in two stages, first utilizing the accept/reject classifier and then further annotating accepted particles with the intact/permeable classifier and the rejected particles with the debris/aggregate.

**Cell culture**. A549, PC9, MDA-MB-361, PA-TU-8902, HT-29, NCI-H1666, NCI-H2228, MM.1S, MM.1R, H929, U266, and KMS-12-PE cells were maintained in a base media supplemented with fetal bovine serum (FBS; Sigma Aldrich, Cat#F4135), antibiotic-antimycotic (Gibco, Cat#15240062) and HEPES (Gibco, Cat#15630080), and stored in a humidified incubator at 37 °C, 5% CO$_2$. Base media was either RPMI1640 + GlutaMAX (Gibco, Cat#61870), DMEM + glucose + GlutaMAX (Gibco, Cat#10566024), or McCoy's 5 A Medium (ATCC, Cat#302007). H929 cells were cultured in media supplemented with 2-Mercaptoethanol (Sigma, Cat#97622). For passage, adherent cell lines were treated with 0.25% trypsin-EDTA (Gibco, Cat#25200) as recommended by the manufacturer. Cell lines were obtained from ATCC, DSMZ, or ECACC, except for KMS-12-PE cells, which were received

as part of a collaboration with the Parekh Lab at MSSM. All cell lines tested negative for mycoplasma on a monthly basis.

**Drug dosing**. Drugs were obtained from MedChemExpress or SelleckChem as lyophilized powders and were reconstituted in an appropriate solvent and stored as single-use aliquots at −80 °C for long-term use. For mass response assays, cells were dosed at $2 \times 10^5$ cells/mL and were dosed at the determined drug concentrations for the specified duration. Control cells were dosed at 0.25% DMSO (vehicle control). Cells were then plated in 6 or 12-well standard polystyrene plates and incubated for the specified period. For suspension cell lines, at the time of measurement, cells were gently resuspended, rinsed with a complete medium, centrifuged for 5 min at $300 \times g$, and resuspended in a medium for measurement. For adherent cell lines cells, at the time of measurement, the supernatant of each well was collected, the well was rinsed with PBS, and cells were detached using 0.25% Trypsin-EDTA (Gibco, Cat#25200) for 7 min. The detached cells were then combined with the supernatant for each well, centrifuged for 5 min at $300 \times g$, and resuspended in the medium for measurement. Isolated primary cells were plated into 24 or 96-well lo-bind plates (Corning: Cat# 3473), and following incubation, cells were gently resuspended, rinsed with complete medium, centrifuged for 5 min at $300 \times g$, and resuspended in a medium for measurement.

**Sample shipment**. Primary cancer specimens were collected from patients upon provision of informed consent and consistent with the IRB-approved protocols (WCMC IRB # 0010004608, ISMMS IRB # STUDY-18-00456, CSHRI IRB # 1738582-3). Once the primary sample was collected, the shipper was sent overnight back to the Travera laboratory for processing. All FNA and malignant fluid samples were shipped in shippers maintaining 4 °C (FedEx, standard duration cooling unit). Fine-needle aspirates were resuspended in HypoThermosol (BioLife Solutions) for preservation during shipment. Blood and bone marrow specimens were placed in green-top sodium-heparin clinical specimen tubes (BD Vacutainer), and shipped in controlled room temperature shippers maintaining temperatures between 15–20 °C (Inmark Life Sciences). Each shipper was otherwise prepared consistent with the regulations for a Category B shipment of biological substances.

**Tumor cell isolation**. Blood and bone marrow samples were processed by upstream filtration through a 70-µm mesh filter (pluriSelect), followed by Ficoll density gradient centrifugation (Sigma Aldrich). Mononuclear cell populations were then subjected to positive selection using CD138+ or CD33+ magnetic microbeads (Miltenyi) and separated using an AutoMACS Pro Separator (Miltenyi) following the manufacturer's protocols. Purified cells were resuspended in RPMI media supplemented with Glutamax and 10% FBS. Fine-needle aspirate samples were first subjected to red blood cell lysis (BD PharmLyse), followed by mechanical and enzymatic digestion using the Miltenyi human tumor dissociation kit following the manufacturer's protocol. Tumor cells were then purified using the Miltenyi human tumor cell isolation kit following the manufacturer's protocol. Pleural effusion samples were similarly first subjected to red blood cell lysis followed by MACS-based tumor cell isolation using the same Miltenyi tumor cell isolation kit. For both FNA and effusion specimens, enriched tumor cells were resuspended in DMEM media with Glutamax and 20% FBS.

**Viability assays**. The viability response of cells was assessed using CellTitre-Glo 2.0 (Promega, Cat#G9242), and luminescence was quantitated by a plate reader. Cells were plated in triplicate in a flat bottom 96-well plate (Corning, Cat#3903) at starting concentrations of either $2 \times 10^4$ or $2 \times 10^5$, depending on cell doubling time. Drug treatments were assessed between 24–144 h. All measurements were performed based on the manufacturer's protocol.

**Flow cytometry assays**. All flow cytometry measurements were performed on a MACSQuant 10 cytometer (Miltenyi Biotec, Auburn, CA). Data analysis was performed with FlowJo (10.8.1). Cell viability was assessed by staining cells with Annexin V (Biolegend, CAT# 640945) and 4′,6-Diamidino-2-Phenylindole, Dilactate (DAPI) (Biolegend, CAT# 422801), viable cells were defined as cells staining negative for both DAPI and Annexin V. To compare flow cytometry data with mass data (Fig. 5 and Supplementary Fig. 9), subsets of 2500 cells were randomly sampled from each flow cytometry data set 1000 times to define a 95% confidence interval for cell viability readouts. These measurements were then compared to the control condition using a 3% limit of decision (described above for mass response measurements) to determine $p$ values. To faithfully compare the magnitude of response between flow cytometry and mass readouts, the y-axis limit for mass response measurements was determined by finding the mass response between image events classified as "Cells" versus "Permeable" within the control population of cells measured for each experiment as a proxy for 100% viability loss as determined by flow cytometry (Supplementary Fig. 4).

**Reporting summary**. Further information on research design is available in the Nature Portfolio Reporting Summary linked to this article.

## Data availability

The datasets generated during and/or analyzed during the current study are available from the corresponding author on reasonable request. All source data for figures can be found in Supplementary Data 1.

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

## Acknowledgements

The authors would like to thank Scott Manalis and Keith Ligon for their feedback and helpful discussions. This work was supported by the Phase I NCI SBIR 1R43CA228872-01A1 and the Phase II NSF SBIR 2026060. Figures were generated using BioRender.com.

## Author contributions

R.J.K., M.M.S., S.O., M.V., A.M., R.L., and C.A.R. defined the research strategy and designed the experiments. R.J.K., A.M., M.V., and R.L. performed experiments. R.J.K., M.M.S., S.O., A.M., M.V., R.L., and M.F. performed data analysis. R.J.K., M.M.S., S.O., and S.C.W. implemented the platform hardware and software. J.F., Z.S., S.S., D.R., D.J., S.A., A.A., J.A.C., R.N., M.R.L., S.P., C.A.R., and A.T. managed clinical specimen collection and shipment including patient identification, patient consenting, sample collection and packaging for shipment, and patient clinical annotation. R.J.K., M.M.S., and S.O. prepared the figures and wrote the manuscript, with feedback from all authors.

## Competing interests

R.J.K., M.M.S., S.O., and C.A.R. are founders of Travera, which is commercializing SMR technology for clinical use. R.J.K., M.M.S., S.O., A.M., M.V., R.L., and C.A.R. are employees of Travera. M.F., S.W., and A.T. receive consulting fees from Travera. The remaining authors declare no competing interests.
