## [Peer Review File · Communications Biology]

Reviewers' comments:

Reviewer #1 (Remarks to the Author):

The utility of Suspended Microchannel Resonators (SMR) and the Mass Accretion Rate measurements for drug screening have been well established in the last decade. Indeed some of the co-authors in this study are also the co-authors in these earlier work. This manuscript extends these techniques greatly in the biological/clinical domain. Specifically, there are three significant advances over the previous work:

- 1) The detailed explanation of a practical pipeline from sample collection to data analysis,
- 2) Testing 60 different types of drug/cell line combinations, as well as 4 types of specimen collection which advances the earlier reports greatly,
- 3) The use of convolutional neural networks (CNN) for real-time labeling of data.

As such, this is a remarkable paper that serves as a bridge between an emerging technology and practical problems in the actual application. For instance number (3), CNN may at first appear as a simplistic feature; however it solves a very important, practical problem. Avoiding cell clusters in microfluidic channels without compromising the throughput or cell viability is difficult; and the real-time CNN can efficiently help in labeling such clusters, and other debris, to stratify the data analysis as shown in Figure 3.

The authors then use these 'curated mass measurements' to first demonstrated the mechanism of action of different types of drugs on the mass signal; tested numerous drugs on cancer cells; and compared the predictive potential of SMR measurements with respect to established techniques.

Overall the experiments and data analyses were carried out diligently and the results are significant. For these reasons, I strongly endorse the publication of this manuscript.

-

I have only one major concern:

- 1) One of the main claims of the paper is that the SMR measurements can predict the efficacy of drugs much earlier than flow cytometry; and Figure 5D is one of the major figure to base these claims. However, in this figure while the SMR data sets have error bars; the flow cytometry (DAPI+/Annexin V+) results do not have such error bars. Although the increase in the flow cytometry signal seems small (given that the percentage of cells are being reported), still to make a proper comparison with the two data sets, one needs to know the error levels which is missing for the flow cytometry experiments.

Minor points

- 2) DMSO treatment on cells have been used in many of the experiments as a form of control run; however, the rationale behind the use of DMSO (and the small but non-negligible shift it induced) was not discussed in the text.

- 3) Is there any insight on the origin of the nonlinear response in Figure 6A ?

Reviewer #2 (Remarks to the Author):

The authors report a cancer drug screening platform for functional precision medicine, based on high-throughput single cell mass sensing complemented with imaging classification. The authors clearly show that the platform can detect the efficacy of cancer drugs as the changes in the mass distribution of the drug-treated cell population. Carefully designed experiments and detailed statistical analysis are presented to show the validity of the measurements.

Overall, the developed platform in this study has a very strong potential to become a practical solution to enable the drug screening for cancer patients and the manuscript would be of great interests to those clinicians and scientists in the field of oncology, drug-discovery, and other areas.

To further strengthen the impact of this study, I would like to suggest followings. First, how many different conditions can be screened by this platform with a single biopsy? It would be difficult to give a simple answer, as it depends on the type of biopsy, the efficiency of cancer cell isolation, and the number of cells for each condition. However, such information would be an important factor to gauge the impact of the developed platform. Second, how the cells (especially adherent cells) were treated with drugs? Were they attached, treated with drugs, and detached again with trypsin? I can't find the detailed protocols for this process, and it would be helpful to provide them in the supplementary information.

Reviewer #3 (Remarks to the Author):

In this Manuscript, Kimmerling et al. have presented a pipeline for drug sensitivity testing of tumor cells using cell mass. The presented approach is built around the proven suspended microchannel resonator technique from Manalis lab. This article delves explicitly into the process of creating a clinical drug testing assay and presents patient results from fine needle aspirates, blood, and pleural samples. The manuscript is very well written, and the presented work definitely has high clinical and academic potential. I recommend the publication of this manuscript, provided the authors can address the following comments:

1) The strength of this technology is the ability to measure subtle changes in the cell populations in response to the drug. This is illustrated in Figure 5D, where dose response as measured by cell mass shows better fidelity than cytometry-based measurement of cell death and apoptosis using DAPI and Annexin V. It is not very clear in this diagram if the authors are presenting the "DAPI+ and Annexin+" cells or "Dapi+ or Annexin+" cells. Given the terminology used, it seems that they are presenting DAPI+ and Annexin+ cells, which will be the necrotic cell population. Authors should instead use the Apoptotic+Necrotic ("Dapi+ or Annexin+") cell populations as a measure of the drug response.

2) It is not entirely convincing that cell mass is the most sensitive method for measuring the drug response. Brightfield imaging and fluorophore-based assessment of cell cycle in conjugation with apoptosis can sensitively profile the three different mechanisms of action (MOA) outlined in Figure 4A.

3) During analysis, the authors reject aggregates of cells. This could lead to misleading results as tumor cells often survive drugs or environmental stresses in groups of cells. By rejecting that population, this technique is rejecting the most vital cellular fraction and, in my mind, the most significant disadvantage of this method. Authors should include aggregates of cells in their analysis.

4) Often, pleural fluids, bone marrow, and fine needle aspirates have various contaminating cells. Despite the most efficient isolation methods, depending on the frequency of tumor cells, post-isolation, their purity will be 10% to >90%. This could be very problematic, as mass measurements will collect data from both the non-depleted contaminating cell population as well as actual tumor cells. It can skew results and lead to misleading data. This approach will work well when tumor cell purity is >90%, but that would not be possible for a considerable number of these samples. Authors should discuss this aspect.

5) Blood samples and pleural fluids can degrade rapidly during overnight shipping, causing a significant number of cells to be apoptotic. Authors should discuss how this degradation would affect their results and analysis.

Response to reviewers' comments on "A pipeline for malignancy and therapy agnostic assessment of cancer drug response using cell mass measurements" by Kimmerling et al.

We thank the Reviewers for their careful consideration and thoughtful comments. We were pleased to see that all three Reviewers found that the work was completed rigorously and highlighted the high potential clinical impact of the platform. Each Reviewer provided helpful suggestions on how to clarify and bolster our results. We very much appreciate this feedback and believe that addressing these highlighted points has strengthened the work significantly.

Below, we address each of the Reviewer's comments (bold blue italics) with a full point-by-point response (plain text), explaining where and how we have revised our manuscript to address all outstanding concerns.

Reviewers' comments:

Reviewer #1 (Remarks to the Author):

The utility of Suspended Microchannel Resonators (SMR) and the Mass Accretion Rate measurements for drug screening have been well established in the last decade. Indeed some of the co-authors in this study are also the co-authors in these earlier work. This manuscript extends these techniques greatly in the biological/clinical domain. Specifically, there are three significant advances over the previous work:

- 1) The detailed explanation of a practical pipeline from sample collection to data analysis,*
- 2) Testing 60 different types of drug/cell line combinations, as well as 4 types of specimen collection which advances the earlier reports greatly,*
- 3) The use of convolutional neural networks (CNN) for real-time labeling of data.*

As such, this is a remarkable paper that serves as a bridge between an emerging technology and practical problems in the actual application. For instance number (3), CNN may at first appear as a simplistic feature; however it solves a very important, practical problem. Avoiding cell clusters in microfluidic channels without compromising the throughput or cell viability is difficult; and the real-time CNN can efficiently help in labeling such clusters, and other debris, to stratify the data analysis as shown in Figure 3.

The authors then use these 'curated mass measurements' to first demonstrated the mechanism of action of different types of drugs on the mass signal; tested numerous drugs on cancer cells; and compared the predictive potential of SMR measurements with respect to established techniques.

Overall the experiments and data analyses were carried out diligently and the results are significant. For these reasons, I strongly endorse the publication of this manuscript.

-

I have only one major concern:

1) One of the main claims of the paper is that the SMR measurements can predict the efficacy of drugs much earlier than flow cytometry; and Figure 5D is one of the major figure to base these claims. However, in this figure while the SMR data sets have error bars; the flow cytometry (DAPI+/Annexin V+) results do not have such error bars. Although the increase in the flow cytometry signal seems small (given that the percentage of cells are being reported), still to make a proper comparison with the two data sets, one needs to know the error levels which is missing for the flow cytometry experiments.

To present a more direct comparison of flow cytometry and mass response measurements we have performed a similar data sampling approach for the flow cytometry data as we performed for the mass response data. Specifically, we collected random subsets of 2,500 cells 1,000 times from the full flow cytometry data sets to define the 95% confidence interval for these readouts. We then performed a similar limit of decision-based statistical comparison as is used for the mass response measurements to define a p-value for each condition tested. We found that there were in fact statistically significant changes in viability for many conditions as measured by flow cytometry. However, the primary argument we are presenting here is that the magnitude of these viability changes for slow acting drugs is low, and insufficient to define an accurate IC_{50} , despite a clear inflection point in the mass responses for a similar set of drug doses tested. We have updated the text to make this point more clearly (lines 286-292) and incorporated a description of the statistical approach used for flow cytometry in the Methods section, along with a description of how we define the y-axis limits for mass response to make a meaningful comparison between flow cytometry and mass data (lines 579-587).

Minor points

2) DMSO treatment on cells have been used in many of the experiments as a form of control run; however, the rationale behind the use of DMSO (and the small but non-negligible shift it induced) was not discussed in the text.

Because most of the drugs we test are dissolved in DMSO prior to dosing, we use 0.25% DMSO treatment as the vehicle control condition for our mass response assay. We have added text in the mass interpretation section that clarifies this point and included a pointer to Supplementary Figure 1 which shows that DMSO has no significant effect on cell mass when compared to untreated cells (lines 146-149).

3) Is there any insight on the origin of the nonlinear response in Figure 6A ?

In the original manuscript, we mistakenly utilized an x axis with evenly spaced ticks despite unequal changes in the fraction of treated cells which gave the impression of a non-linear response. We have corrected Figure 6a to have x axis ticks spaced appropriately for the fractions of treated cells which now shows the expected linearity of the response.

Reviewer #2 (Remarks to the Author):

The authors report a cancer drug screening platform for functional precision medicine, based on high-throughput single cell mass sensing complemented with imaging classification. The authors clearly show that the platform can detect the efficacy of cancer drugs as the changes in the mass distribution of the drug-treated cell population. Carefully designed experiments and detailed statistical analysis are presented to show the validity of the measurements.

Overall, the developed platform in this study has a very strong potential to become a practical solution to enable the drug screening for cancer patients and the manuscript would be of great interests to those clinicians and scientists in the field of oncology, drug-discovery, and other areas.

To further strengthen the impact of this study, I would like to suggest followings.

1) First, how many different conditions can be screened by this platform with a single biopsy? It would be difficult to give a simple answer, as it depends on the type of biopsy, the efficiency of cancer cell isolation, and the number of cells for each condition. However, such information would be an important factor to gauge the impact of the developed platform.

We have included an additional set of mass response measurements that were collected for an acute myelogenous leukemia specimen that had a sufficiently high cell yield to measure a total of 26 replicate conditions across 11 different drug conditions (Supplementary Figure 11). This specimen serves as a contrast to the FNA specimens presented in Figure 7, where cell yield limited the number of drug tests possible, and we believe helps to contextualize the variability of the number of conditions that can be screened for a given sample. We have also included a description of this variability in the discussion to further highlight this point (lines 425-430).

2) Second, how the cells (especially adherent cells) were treated with drugs? Were they attached, treated with drugs, and detached again with trypsin? I can't find the detailed protocols for this process, and it would be helpful to provide them in the supplementary information.

We have added a more thorough description of the cell plating and preparation approaches (including for adherent cells) to the methods section (lines 542-550).

Reviewer #3 (Remarks to the Author):

In this Manuscript, Kimmerling et al. have presented a pipeline for drug sensitivity testing of tumor cells using cell mass. The presented approach is built around the proven suspended microchannel resonator technique from Manalis lab. This article delves explicitly into the process of creating a clinical drug testing assay and presents patient results from fine needle aspirates, blood, and pleural samples. The manuscript is very well written, and the presented work definitely has high clinical and academic potential. I recommend the publication of this manuscript, provided the authors can address

the following comments:

1) The strength of this technology is the ability to measure subtle changes in the cell populations in response to the drug. This is illustrated in Figure 5D, where dose response as measured by cell mass shows better fidelity than cytometry-based measurement of cell death and apoptosis using DAPI and Annexin V. It is not very clear in this diagram if the authors are presenting the “DAPI+ and Annexin+” cells or “Dapi+ or Annexin+” cells. Given the terminology used, it seems that they are presenting DAPI+ and Annexin+ cells, which will be the necrotic cell population. Authors should instead use the Apoptotic+Necrotic (“Dapi+ or Annexin+”) cell populations as a measure of the drug response.

We agree that the inclusion of both DAPI or Annexin V positive cells (i.e. cells called viable are negative for both DAPI and Annexin V) is most appropriate. These were the criteria used for the initial analysis, but we realize this was not made explicit. We have updated the axis labels in Figure 5 and Supplementary Figure 9 to explicitly state “DAPI+ or Annexin V+” on the flow cytometry viability axis. We have also included a description of how we defined viable cells when analyzing the flow cytometry results in the methods section for additional clarity (lines 579-587).

2) It is not entirely convincing that cell mass is the most sensitive method for measuring the drug response. Brightfield imaging and fluorophore-based assessment of cell cycle in conjugation with apoptosis can sensitively profile the three different mechanisms of action (MOA) outlined in Figure 4A.

We believe that the mass response pipeline presented here offers key advantages over existing approaches, but a head-to-head comparison of overall assay sensitivity is beyond the scope of this work. We would agree that we do not provide sufficient evidence to suggest that cell mass is definitively the most sensitive method of measuring drug response. With this in mind, we have tried to exclude any language that suggests that mass response measurements demonstrate superior sensitivity to other multimodal fluorescent detection approaches. Instead, we highlight the promising features of mass response measurements, including a greater magnitude of response detectable at low drug doses when compared with flow cytometry viability readouts (a claim limited to a small subset of fluorescence-based methods that we feel is substantiated by the results presented). Additionally, we attempt to demonstrate concordance with existing fluorescent approaches, as is seen in the cell cycle assessment comparisons presented in Supplementary Figure 5.

3) During analysis, the authors reject aggregates of cells. This could lead to misleading results as tumor cells often survive drugs or environmental stresses in groups of cells. By rejecting that population, this technique is rejecting the most vital cellular fraction and, in my mind, the most significant disadvantage of this method. Authors should include aggregates of cells in their analysis.

Across all the data presented in this manuscript we found that, on average, only ~7% of particles measured were classified as ‘Aggregates’ based on our image classification model. However, exclusion of these events led to a >10% improvement in sampling error (and therefore an improved ability to identify drug response). Given this outsized contribution to measurement noise, and limited fraction of

events, we felt it most prudent to remove 'Aggregate' events from the data analysis along with debris events. We have included two additional panels in Supplementary Figure 4 and have updated the main text accordingly to demonstrate this point (lines 207-208).

4) Often, pleural fluids, bone marrow, and fine needle aspirates have various contaminating cells. Despite the most efficient isolation methods, depending on the frequency of tumor cells, post-isolation, their purity will be 10% to >90%. This could be very problematic, as mass measurements will collect data from both the non-depleted contaminating cell population as well as actual tumor cells. It can skew results and lead to misleading data. This approach will work well when tumor cell purity is >90%, but that would not be possible for a considerable number of these samples. Authors should discuss this aspect.

As this platform advances into clinical validation, we agree that managing the complexity of primary specimens, including variability in cellular composition and the loss of cell viability will be critically important for the success of the assay. We have included an overview of these considerations in the discussion (lines 425-430) and have also included Supplementary Note 1 which provides a more detailed discussion of these primary specimen considerations (lines 629-664). We believe this discussion helps to define the challenges of working with primary specimens and lays out the key focuses of future work.

5) Blood samples and pleural fluids can degrade rapidly during overnight shipping, causing a significant number of cells to be apoptotic. Authors should discuss how this degradation would affect their results and analysis.

Considerations of cell viability are also explored in Supplementary Note 1, as described above.

Figure updates

Figure 5C, D: Updated mass measurement plot to include single representative SMR system to compare with flow cytometry and incorporated statistical comparisons and 95% confidence intervals for flow cytometry data (as described in updated methods section, lines 579-587).

Supplementary Figure 4: Added panels B and C to show the fraction of events classified as each particle type (B) and the reduction of internal distance measurements offered by excluding either debris or aggregate events (C) for the image classification data presented in Figure 3C. The reference to these updated panels can be found in lines 207-208 in the manuscript.

Figure 6A: Adjusted the x axis label spacing to appropriately capture the difference between the fractions of drug treated cells being plotted and demonstrate the linearity of response.

Supplementary Figure 9: As with Figures 5C, D -- updated mass measurement plot to include single representative SMR system to compare with flow cytometry and incorporated statistical comparisons and 95% confidence intervals for flow cytometry data (as described in updated methods section, lines 579-587).

Supplementary Figure 11: Added additional primary specimen data for acute myelogenous leukemia specimen to demonstrate the total number of drug conditions that can be screened for a single specimen. These results provide context for the primary specimen discussion presented in lines 425-430.

REVIEWERS' COMMENTS:

Reviewer #1 (Remarks to the Author):

The authors have responded to the concerns raised in the previous peer review step in a satisfactory factor. Therefore, I endorse the publication as is.

Reviewer #2 (Remarks to the Author):

The authors clearly addressed my concerns & questions, and updated the manuscript accordingly. Also, the response to other reviewers' comment seems to be reasonable. I believe this manuscript is good for publication.

Reviewer #3 (Remarks to the Author):

Authors has mostly answered all my comments satisfactorily. I recommend the publication of this manuscript.

As a note: I would strongly recommend authors to consider aggregates of cells, even if it is 7% of all the cells as this may be the most important cell population from survival point of view.

Response to reviewers' final comments on "A pipeline for malignancy and therapy agnostic assessment of cancer drug response using cell mass measurements" by Kimmerling et al.

Once again, we thank the reviewers for their constructive comments and suggestions, we are pleased to see that all reviewers feel that we effectively addressed their concerns.

Below, we address Reviewer #3's final comment (plaint text).

Reviewer #1 (Remarks to the Author):

The authors have responded to the concerns raised in the previous peer review step in a satisfactory factor. Therefore, I endorse the publication as is.

Reviewer #2 (Remarks to the Author):

The authors clearly addressed my concerns & questions, and updated the manuscript accordingly. Also, the response to other reviewers' comment seems to be reasonable. I believe this manuscript is good for publication.

Reviewer #3 (Remarks to the Author):

Authors has mostly answered all my comments satisfactorily. I recommend the publication of this manuscript.

As a note: I would strongly recommend authors to consider aggregates of cells, even if it is 7% of all the cells as this may be the most important cell population from survival point of view.

While we appreciate that cellular aggregates may represent a biologically significant population, we do not believe that our platform is well suited to including these events in the interpretation of single-cell drug response. Because our statistical approach relies on the direct comparison of masses between events accepted after image classification, the inclusion of aggregate events would limit the ability to distinguish changes in cell mass from changes in cellular aggregation. While such aggregation readouts could potentially offer meaningful information about drug response, this type of approach is beyond the scope of the workflow presented here. We have included an additional discussion of aggregate events in Supplementary Figure 4 to help contextualize this limitation for readers.